# Evaluation of the Susceptibility of Lotus Seeds (*Nelumbo nucifera* Gaertn.) to *Aspergillus flavus* Infection and Aflatoxin Contamination

**DOI:** 10.3390/toxins16010029

**Published:** 2024-01-07

**Authors:** Abdelrahman Elamin, Sharmin Sultana, Shohei Sakuda

**Affiliations:** Department of Biosciences, Teikyo University, 1-1 Toyosatodai, Utsunomiya 320-8551, Japan; a.elamin@nasu.bio.teikyo-u.ac.jp (A.E.); sharmin19822011@gmail.com (S.S.)

**Keywords:** aflatoxins, lotus seed, water-gap, HPLC, LM, SEM

## Abstract

The seeds of lotus (*Nelumbo nucifera* Gaertn.) have been used as significant medicinal and nutritional ingredients worldwide. The abundant proteins and polysaccharides in lotus seeds make them susceptible to contamination by aflatoxin (AF), a fungal toxic metabolite. This study was conducted to investigate the susceptibility of lotus seeds at different stages of ripening to AF contamination, as well as the mechanism of the contamination. Seven groups of lotus receptacles with seeds at different ripening stages (A–G, from immature to mature) were used for the experiment. Spores of *Aspergillus flavus*, an AF producer, were inoculated on the water-gap area of the seeds in each receptacle. Then, each receptacle was covered with a sterilized bag, and its stalk part was soaked in water containing a life-prolonging agent, after which it was kept at room temperature for 14 days. The AF content of each whole inoculated seed from the A–G groups and that of each seed part (pericarp, cotyledon, and embryo) from the D and E groups were determined using high-performance liquid chromatography. Microtome sections were prepared from the samples and observed under a light microscope and scanning electron microscope. The seeds from the A and D groups had higher AF contents than the seeds from the B, C, E, F, and G groups, indicating that the condition of the water-gap area and the development of the embryo and cotyledon parts of the seeds are associated with AF contamination.

## 1. Introduction

Lotus (*Nelumbo nucifera* Gaertn.) belongs to the family Nelumbonaceae, which has a relatively wide geographical distribution and biological diversity [1]. All parts of the lotus plant are used as food and medicine. Lotus seeds have been widely used in the world because of their abundant content of major food components (lipids, proteins, starch, vitamins, and minerals) and bioactive compounds. The seed contains 61–62% carbohydrates, 16–21% total protein, 2.4–3% crude fat, and 5–9% moisture [2]. The seed also contains abundant amounts of health-promoting phytochemicals that have high nutritional and pharmaceutical value [3,4]. In traditional Chinese medicine, *N. nucifera* seeds are used for the treatment of cancer, tissue inflammation, and skin disorders, as well as being antiemetic and diuretic agents and thermoregulators for reducing body temperature [5].

Nevertheless, the abundant amounts of proteins and polysaccharides in lotus seeds result in their susceptibility to infection with toxigenic fungi in the preharvest, postharvest, processing, storage, and transportation processes under appropriate humidity and temperature conditions [2,6], especially during the rainy season, leading to the production and residue of various mycotoxins [7].

Mycotoxins are a group of approximately 400 toxic secondary metabolites produced by fungi such as *Aspergillus*, *Penicillium*, *Fusarium*, *Claviceps*, and *Alternaria* [8]. These naturally occurring compounds are found in a wide range of agricultural products and medicinal herbs in various regions of the world [9]. Aflatoxins (AFs), the most prevalent mycotoxins that contaminate foods, are produced primarily by *Aspergillus flavus* and *A. parasiticus* and comprise a group involving the four major AFs: aflatoxin B_1_ (AFB_1_), aflatoxin B_2_ (AFB_2_), aflatoxin G_1_ (AFG_1_), and aflatoxin G_2_ (AFG_2_) [10]. They are carcinogenic compounds classified as group 1 carcinogens to humans by the International Agency for Research on Cancer (IARC) [11].

A study reported that 95% of the batches of lotus seeds collected from different drug stores or markets in China were contaminated with AFs at levels ranging from 0.02 to 688.4 μg/kg [12]. The contamination of medicinal herbs used in the pharmaceutical industry with mycotoxins represents one of the problems that trouble researchers in this field, and requires urgent solutions. Therefore, the determination of the mechanism of AF contamination in lotus seeds could represent one of the solutions to prevent the contamination of these herbs. However, there is limited information on the mechanism of contamination, such as how fungi penetrate the seed and which ripening stage of the seed is susceptible to the contamination.

Microscopic techniques are used to determine the quality of herbal drugs. Light microscopy (LM) and scanning electron microscopy (SEM) have been previously used successfully to determine the penetration paths and accumulation of fungal mycelia inside medicinal herbs [13].

The aims of the present study were to evaluate (1) the susceptibility of whole seed to AF contamination based on maturity, (2) the highly susceptible part of the seed to AF contamination, and (3) the paths of fungal mycelial penetration to the inner region of the lotus seed. High-performance liquid chromatography (HPLC) was used to quantify the AFs in whole lotus seeds and in each of their parts separately. Longitudinal sections of seeds at different stages of maturity were examined using LM to monitor the changes in the structure of the water-gap region during maturation. Sections of contaminated lotus seeds at different stages of maturity were examined using SEM to determine the paths by which the fungi had entered the samples.

## 2. Results

### 2.1. Susceptibility of Lotus Seeds to AF Contamination

#### 2.1.1. Susceptibility of Whole Seeds at Different Maturity Stages to AF Contamination

Lotus seeds in seven receptacle groups, which were classified based on maturity (A–G) (Figure 1), were inoculated with *A. flavus* spores and incubated for 14 days. HPLC analysis revealed higher AF concentrations (total of AFB_1_ and AFB_2_ concentrations) in the whole seeds of the A and D groups than in the seeds of the B, C, E, F, and G groups (Figure 2). A significant difference was observed in the AF concentrations between the seeds of the A group and those of the B, C, E, F, and G groups.

#### 2.1.2. Susceptibility of Different Parts of Lotus Seeds to AF Contamination

The seeds at mid-mature stages (D and E) were selected to evaluate the susceptibility of the different parts of the lotus seeds to AF contamination because of the following two reasons: (1) the embryo part of the seeds at these stages has grown to such an extent that its susceptibility to AF contamination can be easily compared with that of the other two parts (pericarp and cotyledon) in contrast to the seeds at immature stages (A, B, and C), and (2) the high susceptibility of whole seeds at the D stage to AF contamination, as described in Section 2.1.1. After inoculating *A. flavus* spores on the seeds in the receptacles of the D and E groups and incubating for 14 days, each seed was separated into three parts (pericarp (Pe), cotyledon (Co), and embryo (Em), as shown in Figure 3), and the AFs in each part were quantified. The pericarp part contains the base of the style (St), multiple layers of epidermal mesophyll cells (Ep), palisade cells (Pa), and protuberance cavity (Pr). The seedcoat (Se) is involved in the cotyledon part. The AF concentration in the embryo part of the seeds of the D group was significantly higher than that in the pericarp part. The AF concentration in the embryo part of the seeds of the E group was significantly higher than that in the cotyledon and pericarp parts (Figure 4).

### 2.2. Changes in the Water-Gap Structure during Maturation

The term “water-gap” is used to define the specialized anatomical structure in the palisade layers that break the physical dormancy of the seed and allow water to enter the seed, which is essential for water uptake to the embryo and subsequent germination under natural conditions [14,15,16]. Therefore, it could be considered a potential path of entry for fungi to the inner part of the seed. The structure of the water-gap region at the protuberance organ (Pr) (Figure 3) of the lotus seed and its improvement during maturation were investigated by LM. The general form of the protuberance organ was found at all stages of maturity, showing its outer cavity (Oc) and inner cavity (Ic) (Figure 5a–d). The sclerenchyma cells of the wall of the protuberance organ (Sc), which have a thick, lignified secondary cell wall and are dead at maturity, were not observed in the immature stages (A and B) (Figure 5a,b), but they were clearly visible in the mid-mature and mature stages (D and G) (Figure 5c,d). The crystalliferous cells (Lc), which outline the protuberance organ with modified sclerenchyma cells and enhance the protective role of the pericarp, were observed only at the protuberance organ of the G-stage seed (Figure 5d).

### 2.3. Path of Fungal Mycelial Penetration of Lotus Seeds

To demonstrate the path of fungal mycelial penetration, sections from parts of the lotus seeds at different maturity stages were investigated using SEM. The results showed that the fungal mycelia penetrated the inner part of the lotus seeds through the water-gap region at the protuberance organ (Figure 6). The intensive invasion of the fungal mycelia to the water-gap region of the seeds at immature stages (A and B) was verified (Figure 6f,j). The fungal mycelia accumulated on the cotyledons of the seeds at the A and B maturity stages (Figure 6g,h,k,l). The results also showed fungal mycelial penetration from the water-gap region of the seeds at mid-mature stages (D and E) (Figure 6n,r). The contamination of the cotyledon part of seeds at the D and E stages was also recorded, but it was less than that observed for the immature seeds (Figure 6o,s). In contrast to that observed for immature and mid-mature seeds, no penetration to the inner region of the mature seeds (G stage) was observed (Figure 6v,w).

## 3. Discussion

One of the reasons why AFs are one of the most challenging mycotoxins is the fact that they can be produced by aflatoxigenic fungi not only at postharvest stages, including storage, but also at the preharvest stage [17]. Lotus seeds are commonly infected in the field (preharvest) with various fungi, which may cause AF contamination. *A. flavus* was reported as one of the fungi isolated from lotus seeds harvested in China [18]. It was demonstrated that the water-gap region in the cavity of a protuberance is the only permeable region in the lotus seed [19]. In the present study, we simulated the AF contamination of the lotus seed in the preharvest stage. Lotus receptacles with seeds at different maturity stages were collected from the field, and their stalks were immersed in a life-prolonging solution. The water-gap region of the seed was inoculated with *A. flavus* spores, and each receptacle was covered with a sterilized bag to avoid contamination with any other microorganism and incubated for 14 days at room temperature. Our simulations confirmed the previously mentioned possibility of preharvest contamination with *A. flavus* and AFs.

Although many studies have examined lotus seeds, which will be mentioned later, in terms of structure, water content, etc., none of these studies have dealt with tracking the relationship between the maturity of the seeds and their susceptibility to aflatoxin contamination. In the present study, the AF concentration in the seeds from the A and D groups was higher than that in the seeds from the B, C, E, F, and G groups, suggesting that the seeds at the early immature and mid-mature stages were highly susceptible to AF contamination. The higher AF concentration in the A group and the lower AF concentration in the G group might correlate with the moisture content of seeds and the fungal permeability through the water-gap, as mentioned in previous studies [20,21]. During the continuous decrease in moisture content by maturation, seeds will achieve the moisture level for the initiation of impermeability. Seeds with this moisture content are in a transition state, and, thus, the subsequent maintenance or loss of impermeability depends on the relative humidity to which these seeds are exposed. Despite the true connection between environmental stresses, such as moisture content and microbial attack, and permeability [22], it appears that this is not the only factor controlling this mechanism and there are other influential factors that cannot be ignored. The seed structure also develops and can play a vital role in reducing the permeability of the seed to fungal invasion. The pericarp is the first line of defense in protecting lotus fruits from bacterial and fungal penetration. Exudates from the lotus pericarp appear to prevent the growth of certain fungi [23,24]. In the immature stages, the lotus seed is covered in a green, soft, and easily peelable pericarp containing a moist and soft cotyledon and the developing embryo, in contrast to the seed at the mature stage, whose pericarp turns dark brown and hardens, and both the cotyledon and embryo become considerably dry [25,26]. Since the water-gap area is the only permeable area in lotus seeds [19], it was important to study the change in its structure with maturity and the extent of its effect on the ability of the fungi to penetrate the seed. In the present study, the microscopic examination focused on the relationship between the development of the protuberance organ (water-gap region) during the ripening process and its role in reducing fungal mycelial and AF contamination. The presence of the sclerenchyma cells with a thick and lignified secondary cell wall in addition to crystalliferous cells in the seeds of the mature stage compared with other stages (Figure 5) could be the reason for the cease of the fungal penetration and AF contamination of the seeds at this stage. The higher AF concentration in the lotus seeds at the mid-mature stage than that in other seeds is consistent with other reports suggesting that the concentrations of protein, soluble sugar, amino acids, and fatty acids at the mid-mature stages of medicinal herbs increase the susceptibility of the fruit to AF contamination [27,28]. A previous study investigated the accumulation of starch and proteins in the cotyledon of lotus seeds at different stages of maturity [29]. No accumulation of starch and proteins was observed in the immature seeds, whereas starch and proteins accumulated rapidly in mid-mature seeds, and the accumulation rate was slow in mature seeds. Among the three parts of the seeds of the D and E groups, the AF concentration was the highest in the embryo part. The high susceptibility of the embryo, the nutrient-rich part [30,31], of the lotus seed to fungal invasion and high AF concentration was similar to that observed in a previous jujube seed experiment [22]. The high accumulation of nutrients in the embryo, especially at the mid-mature stage, might stimulate higher AF contamination in the embryo than in other parts of the seed [20].

Mycotoxin control and prevention include several strategies: blocking the infection process, biological control, and managing environmental factors [32]. One of these strategies could not work alone to cease all infection pathways. In our research project, we approached control from a preventive perspective by determining which stages of growth and which parts of the medicinal herbs are most susceptible to contamination with mycotoxins, in addition to the mechanism of contamination. Accurate knowledge of the nature of medicinal herbs, their sensitivity to infection, and which types of fungi and mycotoxins are more likely to contaminate the herb makes those working in the field of herbal medicine capable of providing proactive protection.

## 4. Conclusions

The susceptibility of lotus seeds to AF contamination varies according to the maturity stage. The HPLC analysis revealed a high susceptibility of immature and mid-mature seeds to AF contamination in contrast to mature seeds (G). The embryo was the part of the lotus seed most susceptible to AF contamination. The water-gap region at the protuberance organ was confirmed to be the path of fungal mycelial penetration to the inner parts of the seed. The changes occurring in the water content and seed structure, especially the protuberance organ, due to maturation may play a vital role in the cessation of fungal mycelial penetration to the inner seed parts, thus reducing the AF concentration in the mature seed.

The assessment of AF distribution among the different parts of the lotus seed using imaging techniques is an area that requires further attention in order to determine the infection mechanisms.

## 5. Materials and Methods

### 5.1. Sample Preparation

Lotus plants grown in Ibaraki Prefecture, Japan, were used in this experiment. After harvesting, a solution of water:a life-prolonging agent (Chrysal Japan Co., Ltd., Osaka, Japan) (50:1 *v*/*v*) was poured onto lotus receptacles through the stems to keep them alive till the beginning of the experiment. The receptacles were classified into seven groups (A–G) based on the maturity stage, with three receptacles in each group as follows: groups A, B, and C (immature stages); groups D and E (mid-mature stages); and groups F and G (mature stages). One receptacle in each group was used as a control to confirm whether there was any contamination with the aflatoxigenic strain before conducting the experiment. The other two receptacles were used for artificial contamination experiments. The seeds of the receptacles of each group used for the artificial contamination experiment were inoculated with 10 µL of the spore suspension (4.6 × 10^6^/mL) of *A. flavus* IFM 47798 on the water-gap area of each seed. Each receptacle containing the inoculated seeds or control seeds was covered with a sterilized bag (Hogy Medical Co., Ltd., Tokyo, Japan), and its stalk part was soaked in 300 mL of a solution of water:a life-prolonging agent (50:1 *v*/*v*) in a 500 mL conical flask and then incubated for 14 days at room temperature. The water:life-prolonging agent solution in each conical flask was changed every day. The bottom part of the receptacle stalk was cut by 1 cm every 2 days. After 14 days of incubation, some of the inoculated seeds of each group that were of a similar size were taken from the receptacles, soaked in ethanol:water (1:1 *v*/*v*) for 30 min, and desiccated in an oven (50 °C). The collected seeds from the A–G groups (*n* = 12, 15, 17, 16, 17, 10, and 14, respectively) were used for analyzing the AF concentration.

For the experiment described in Section 2.3, three seeds were collected from each of the control receptacles of the seven groups to perform artificial contamination (Appendix A). The aim of this experiment was to obtain seeds with high AF contamination through which the paths of fungal mycelial penetration to the inner parts of the seeds can be revealed. A sucrose-free Czapek–Dox agar (CZA) medium was prepared according to a previously reported method [33]. The seeds from each group were soaked separately in an ethanol:water solution (1:1, *v*/*v*) for 30 min and desiccated in an oven at 50 °C. Next, they were placed on the surface of CZA medium in Petri dishes (150 mm × 25 mm), which were spread with 250 µL of the spore suspension (4.6 × 10^6^/mL) of *A. flavus* IFM 47798. The Petri dishes were incubated at 25 °C for 14 days, after which the seeds from each group were washed with ethanol:water (1:1, *v*/*v*), desiccated in an oven at 50 °C, and examined using SEM.

### 5.2. AF Quantification using HPLC

Each whole seed from the A, B, C, F, and G groups and each separate part (pericarp, cotyledon, and embryo) from the D and E groups were ground (Wonder Crusher WC-3; Osaka Chemical Co., Ltd., Osaka, Japan), weighed, and mixed with 1 mL of acetonitrile:water:methanol (6:4:1, *v*/*v*/*v*). The mixture was vortexed for 5 min at room temperature. After centrifugation (4770× *g*, 10 min, 4 °C), phosphate-buffered saline (PBS), containing 0.01% Tween 20 (Kanto Chemical Co., Inc., Tokyo, Japan), was added to 0.4 mL of the supernatant to a final volume of 10 mL. The mixture was filtered through a glass-fiber filter paper (GA-100; Advantec Toyo Kaisha, Ltd., Tokyo, Japan) and transferred to an immunoaffinity column (Aflaking, Horiba, Ltd., Kyoto, Japan). Next, the column was washed twice with 3 mL of PBS and 3 mL of water and eluted with acetonitrile (3 mL). The eluate was dried under N_2_ gas and mixed with 0.1 mL of water:acetonitrile (9: 1, *v*/*v*) and 0.1 mL of trifluoroacetic acid, and then vortexed for 15 min. After adding 0.3 mL of water:acetonitrile (9:1, *v*/*v*), the reaction mixture was filtered through a 0.2 µm syringe filter (Minisart^®^ RC 4; Sartorius Stedim Lab. Ltd., Stonehouse, UK) and subjected to HPLC-FLD (Capcell Pak C_18_ UG 120 column, 250 × 4.6 mm inner diameter; Osaka Soda Co., Ltd., Osaka, Japan). An isocratic elution of acetonitrile:methanol:water (1:3:6, *v*/*v*/*v*) over 20 min at a flow rate of 1.0 mL was used for the detection of fluorescence at 365 nm excitation and 450 nm emission. The aflatoxin mixture standard solution (FUJIFILM Wako Chemicals, Osaka, Japan) was used to prepare calibration solutions for HPLC-FLD determinations and recovery experiments. The retention times, limits of detection, and limit of quantification of AFB_1_ and AFB_2_ were 7.15 and 12.60 min, 0.1 and 0.1 µg/kg, and 0.25 and 0.25 µg/kg, respectively.

The data were represented as the average (mean) ± standard deviation (SD). The data were analyzed using one-way ANOVA followed by Tukey’s HSD test for specific comparisons between groups. *p* < 0.05 was regarded as statistically significant. Data analysis was performed using Microsoft Excel version 2310 and IBM SPSS Statistics (trial version) for Windows.

### 5.3. Preparation of Longitudinal Sections

Lotus seeds (A–G groups) were dipped in an embedding medium (Tissue-Tek^®^ O.C.T, Sakura Finetek Japan Co., Ltd., Tokyo, Japan) to bind the seeds with the specimen block. An adhesive film (cryofilm type 2C (9); SECTION-LAB Co., Ltd., Hiroshima, Japan) was attached to the sample cross sections, followed by slicing (20 µm thickness) using a cryostat (CM1860; Leica Microsystems, Wetzlar, Germany). The film was then attached to a glass slide (Superfrost Plus; Fisherbrand, USA) with adhesive tape and stored at −80 °C until investigation [34]. The sections were examined via LM.

### 5.4. LM and SEM

The longitudinal sections of the lotus seeds prepared using a microtome and fixed onto glass slides, as described in Section 5.3, were observed under an Olympus BX53 light microscope (Olympus Corporation, Tokyo, Japan) to monitor the changes in the structure of the water-gap region of the lotus seeds during maturation.

SEM was performed on a Hitachi TM3030 tabletop microscope (Hitachi, Ltd., Tokyo, Japan) using a high vacuum. The images were obtained in backscattered electron image mode. The voltage was 15 kV, while the working distance was 5.00 mm. SEM was used to observe the penetration of fungal mycelia to the inner parts of the lotus seeds using sections from the seeds at different stages of maturity.

### 5.5. Stereomicroscope (SM)

The structure of the lotus seeds was observed under an Olympus SZH Stereomicroscope (Olympus Co., Tokyo, Japan).

## Figures and Tables

**Figure 1 toxins-16-00029-f001:**
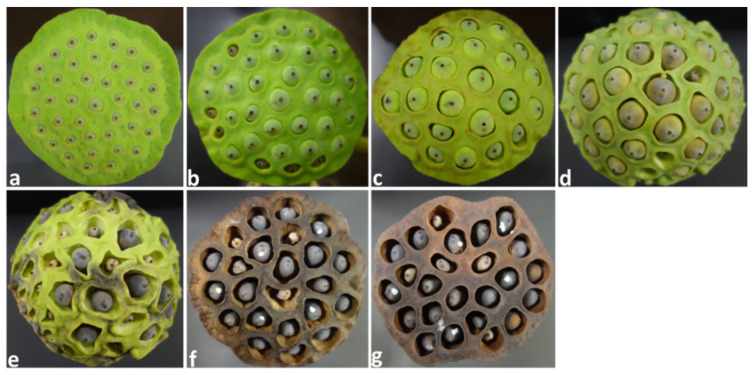
Lotus receptacles with seeds at different maturity stages (A–G). (**a**). A stage; (**b**). B stage; (**c**). C stage; (**d**). D stage; (**e**). E stage; (**f**). F stage; and (**g**). G stage.

**Figure 2 toxins-16-00029-f002:**
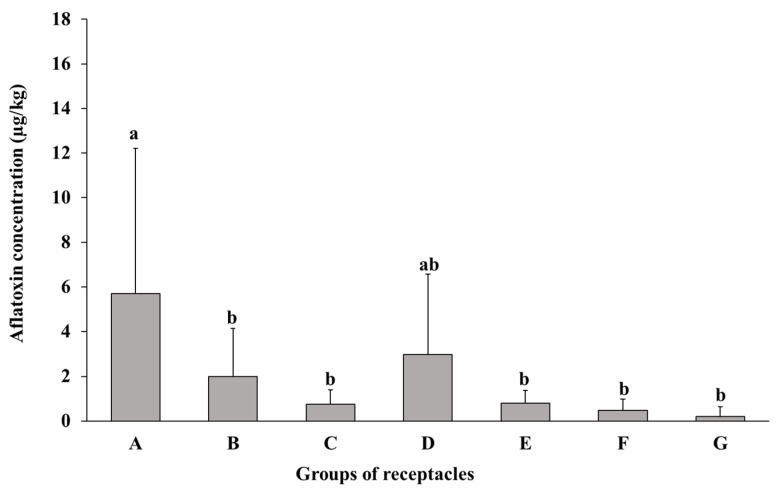
Bar graph showing the mean comparison (±standard deviation) of AF concentrations (total of AFB_1_ and AFB_2_ concentrations) in the seeds inoculated with *A. flavus* from the receptacle groups (A–G) (*n* = 12, 15, 17, 16, 17, 10, and 14, respectively). No contamination was observed in the non-inoculated seeds (*n* > 3 of each stage) from the control receptacles at all stages. Different letters above the bars represent significant differences according to Tukey’s HSD (*p* < 0.05).

**Figure 3 toxins-16-00029-f003:**
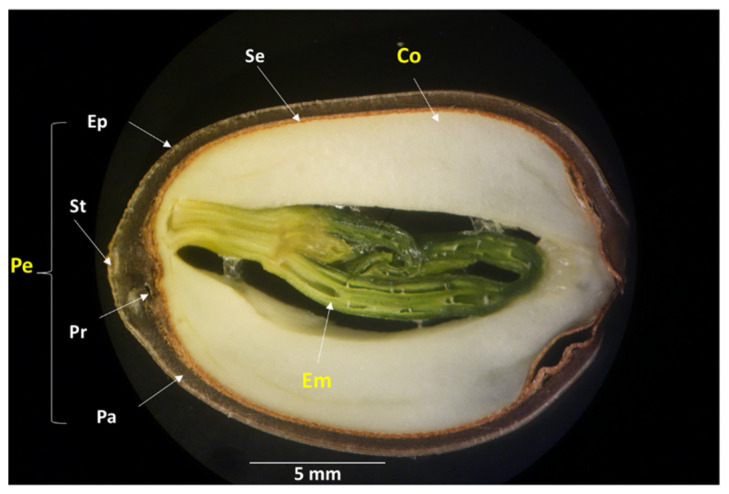
Light micrograph of the longitudinal section of the uncontaminated lotus seed (D stage) showing its structure. Abbreviations: Pe: pericarp; St: base of style; Ep: multiple layers of epidermal mesophyll cells; Pa: palisade cells; Pr: protuberance cavity; Se: seedcoat; Co: cotyledon; Em: embryo.

**Figure 4 toxins-16-00029-f004:**
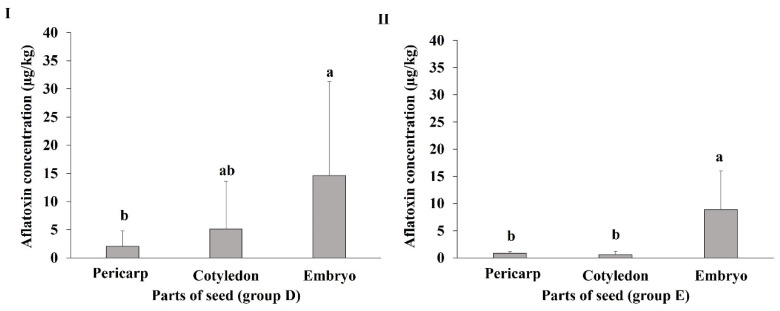
Bar graph showing the mean comparison (±standard deviation) of AF concentrations (total of AFB_1_ and AFB_2_ concentrations) in the pericarp, cotyledon, and embryo of group D (**I**) and E (**II**) seeds (*n* = 11). Different letters above the bars represent significant differences according to Tukey’s HSD (*p* < 0.05).

**Figure 5 toxins-16-00029-f005:**
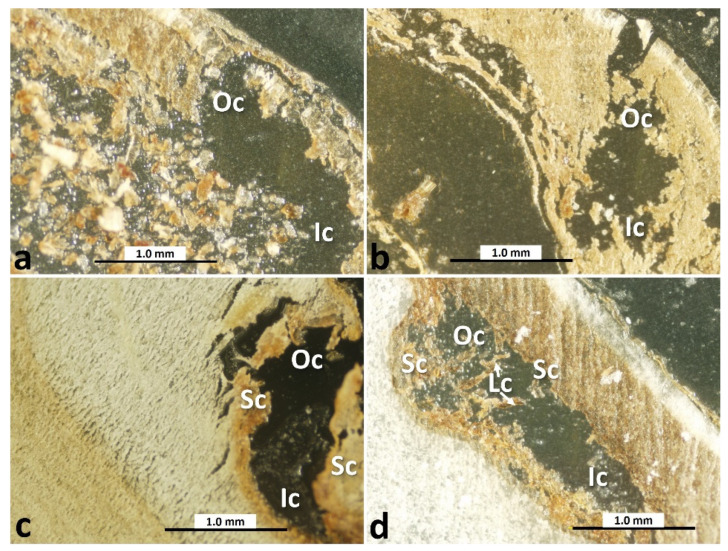
Longitudinal sections of the water-gap region of lotus seeds at different stages of maturity. (**a**). A stage; (**b**). B stage; (**c**). D stage; (**d**). G stage. Abbreviations: Oc, outer cavity of protuberance organ; Ic, inner cavity of protuberance organ; Sc, sclerenchyma cells of wall of protuberance organ; Lc, crystalliferous cells of protuberance organ.

**Figure 6 toxins-16-00029-f006:**
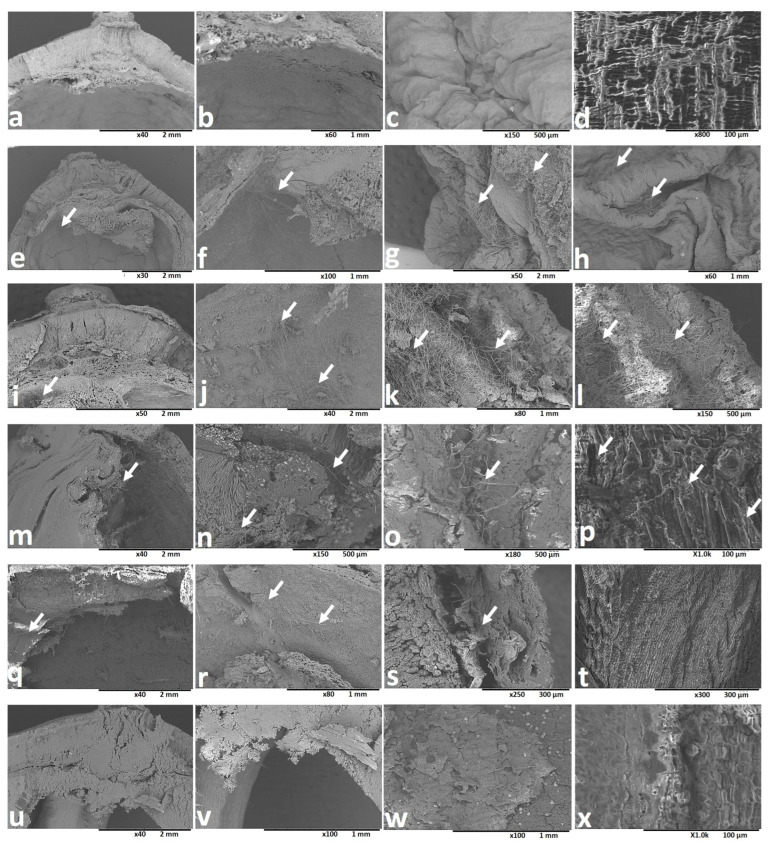
SEM images of the fungal mycelial penetration of lotus seeds at different stages of maturity. (**a**–**d**) control (D stage): (**a**) inner side of pericarp; (**b**) water-gap region; (**c**) cotyledon; (**d**) embryo part. (**e**–**h**) Contaminated seed from A stage: (**e**) inner side of pericarp; (**f**) water-gap region; (**g**,**h**) cotyledon. (**i**–**l**) Contaminated seed from B stage: (**i**) inner side of pericarp; (**j**) water-gap region; (**k**,**l**) cotyledon. (**m**–**p**) Contaminated seed from D stage: (**m**) inner side of pericarp; (**n**) water-gap region; (**o**) cotyledon; (**p**) embryo. (**q**–**t**) Contaminated seed from E stage: (**q**) inner side of pericarp; (**r**) water-gap region; (**s**) cotyledon; (**t**) embryo. (**u**–**x**) Contaminated seed from G stage: (**u**) inner side of pericarp; (**v**) water-gap region; (**w**) cotyledon; (**x**) embryo. White arrows indicate fungal mycelial accumulation.

## Data Availability

The data will be made available on request.

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
