# Peer review of "Evaluation of the Susceptibility of Lotus Seeds (Nelumbo nucifera Gaertn.) to Aspergillus flavus Infection and Aflatoxin Contamination"

_toxins, 2024, doi:10.3390/toxins16010029_

Round 1

Reviewer 1 Report

Comments and Suggestions for Authors

The manuscript entitled on "Evaluation of the susceptibility of lotus seeds (Nelumbo nucif2 era Gaertn.) to aflatoxin contamination" explored that the different part's susceptibility of lotus seeds at different stages of ripening to AF contamination.The research is interesting. Before the manuscript is suggested for publication in the journal, some concerns need to be addressed.

1. Abstract:

The content of the study about "This study was conducted to investigate the susceptibility of lotus seeds at different stages of ripening to AF contamination"was given a partial description .In addition to the different stages of ripening, there are also different parts of the exposure sensitivity and toxin invasion pathways.

2. Introduction

Aflatoxins (AFs) contain more than four species, and these described are not accurately.We suggest to complete it or add the words "and so on".

3. The illustration in Fig 5 ,We suggest that Oc,Ic,Sc, and Lc be labeled with different color letters

In the discussion, the questions raised by the authors are not addressed clearly in the article.Such as "In the present study, the microscopic examination focused on the 196 relationship between the development of the protuberance organ (water-gap region) during the ripening process and its role in reducing fungal mycelial and AF contamination" .The subsequent discussion is not related to the previous question.Please explain more thoroughly.

Reviewer 2 Report

Comments and Suggestions for Authors

    In the present study, the susceptibility of lotus seeds to Aspergillus flavus infection and aflatoxin contamination were determined by inoculation in vivo, HPLC and SEM analyses. The results indicated that immature and mid-mature seeds showed a higher susceptibility, and the embryo was the highest susceptible part to AG contamination. The correlation between susceptibility and structural changes of lotus seeds was demonstrated as well. However, there are some issues to be tackled to improve paper quality.

1.     According to the content, the title should be “Evaluation of the susceptibility of lotus seeds (Nelumbo nucif-2 era Gaertn.) to Aspergillus flavus infection and aflatoxin contamination”.

2.     The significance of this study is not stated in the Introduction or Discussion part.

3.     The description in the section 5.4 is insufficient. Please provide the detailed protocol of SEM.

4.     The introduction of statistical analysis should be supplemented in Materials and Methods.

5.     The product information, peak attributes of AFB1 and AFB2 standard should be described in section 5.2. 

6.     The negative control should be supplemented in Figure 2.

7.     The content related to AF contamination is not enough. It will be better if provide the expression analysis of some key genes involved in AF biosynthesis during the interaction of Aspergillus flavus and lotus seeds at different stages of ripening.

8.     In line 247, the “seven groups . to perform” should be revised.

9.     In Figure 4, The I and II should be replaced by a and b. The distance between letters and error bar should be same for all samples. 

Reviewer 3 Report

Comments and Suggestions for Authors

In the manuscript entitled "Evaluation of the susceptibility of lotus seeds (Nelumbo nucifera Gaertn.) to aflatoxin contamination", authors investigated the susceptibility of lotus seeds (Nelumbo nucifera Gaertn.) at different stages of ripening to contamination by aflatoxin (AF). The experiment involved seven lotus receptacles with seeds at varying ripening stages (labelled A to G, from immature to mature). The seeds in each receptacle were inoculated with spores of Aspergillus flavus at the water-gap area. The analyses were conducted with HPLC. The manuscript is suitable for the aim and scope of the "Toxins" scientific journal; however, it needs some polishing and revision of M&M and discussion for publication in a high-standard journal like Toxins.

Techniques and Analysis: The methodologies used are appropriate for the study's aims. However, a more detailed explanation of the choice of analytical techniques and their limitations would be beneficial. The authors speak about HPLC and, after that, about wavelength for quantifying AFs. However, which detector did you use? DAD? FLD? Did you validate the method? Did you perform control/robustness/repeatability/specificity? Did you use any labelled standard? How can you know this is not interference? Please provide details on the analytical method. I want to see a chromatogram. 

Data Analysis and Interpretation: In the Results section, consider adding statistical analysis details, such as software used for the statistical significance. This would enhance the robustness of the findings. 

Comparative Analysis: In the Discussion section, compare more thoroughly with existing literature. This could involve discussing how your findings align or contrast with previous studies and exploring possible reasons for any discrepancies. I think the data obtained can be considered deeply. The introduction L27-L71 (n=45) is extended as the discussion L167-L212 (n=44). 

Implications and Applications: Expand on the practical applications of your findings in food safety. Discuss potential strategies for mitigating aflatoxin contamination in lotus seeds and other similar crops.

Future Research Directions: Suggest specific areas for future research, such as exploring other methods of aflatoxin detection or mitigation strategies in agricultural practices.

Graphical Data Presentation: Enhance the aesthetics of figures. Consider adding colours to all graphical data. Any idea could be to take the colour of the pictures from Figure 1 and use the hex colour to colour the plot. A boxplot is better than a bar graph. 

Author Response

Please see the attchment.

Round 2

Reviewer 3 Report

Comments and Suggestions for Authors

Authors answered to all my points. No more comments or suggestions needed.

Congrats!